# HYBRID SYMBOLIC-NEURAL MODELS FOR DYNAMICAL SYSTEMS

## ABSTRACT

Dynamical systems are fundamental to modeling the natural world, yet face a persistent trade-off: manually prescribed mechanistic models are interpretable by design but often overly simplistic and misspecified, while flexible data-driven neural methods lack physical insight. Hybrid modeling aims for the best of both worlds by combining a symbolic, physics-based component with a flexible neural network. A critical challenge, however, is that the neural component may re-learn mechanistic parts yielding redundant and uninterpretable models, especially when the symbolic structure itself is discovered from data. Existing methods using standard L2 regularization fail to prevent this overlap in non-convex optimization landscapes created by symbolic regression. We introduce **OrthoReg** (Orthogonal Regularization), an approach that enforces explicit orthogonality between the symbolic and neural components. This guarantees a unique and complementary decomposition preventing the neural component from learning dynamics that can be captured by the symbolic model. We demonstrate empirically on benchmark dynamical systems that OrthoReg improves out-of-distribution generalization, symbolic identification, and sparsity, thereby establishing a new paradigm for building more robust and interpretable hybrid models.

## 1 INTRODUCTION

Dynamical systems modeling has long been a corner stone across the sciences, especially for the natural and life sciences. Applications range from healthcare data Choi et al. (2016); Hess et al. (2024); Seedat et al. (2022), climate modeling Rolnick et al. (2022); Eyring et al. (2024), to power systems Toubeau et al. (2018), to just name a few. However, it faces a fundamental trade-off: symbolic, traditionally manually specified, models provide interpretability by design, but typically not capture complex unknown phenomena; flexible neural networks instead excel at fitting data from dynamical systems Chen et al. (2018) but lack physical insight. Hybrid modeling approaches Rackauckas et al. (2020); Yin et al. (2021); Zou et al. (2024) combine physical priors (predetermined symbolic expressions) with learned neural corrections expected to capture phenomena that are unknown or too complex to model directly. They promise the best of both worlds, but still require substantial prior knowledge in crafting the mechanistic part. In this work, we tackle the problem of discovering mechanistic components from data within a flexible pre-specified function class via dynamic symbolic regression Brunton et al. (2016); Podina et al. (2023); Becker et al. (2023); d'Ascoli et al. (2024), while also capturing residual dynamics outside that function class and explicitly ensure orthogonality, i.e., no redundancy, of the two components.

In their landmark paper, Yin et al. (2021) present the APHYNITY framework, the state-of-the-art in hybrid dynamical systems modeling when the symbolic structure (but not exact parameter values) is known a priori. APHYNITY decomposes the (autonomous) vector field of an ordinary differential equation (ODE) as $f = f_{\mathrm{phy}} + f_{\mathrm{aug}}$, where $f_{\mathrm{phy}} \in \mathcal{F}_{\mathrm{phy}} = \mathrm{span}\{\phi_j\}_{j=1}^k$ captures dynamics within a predetermined library of "symbolic" functions $\phi_j$ (e.g., polynomials, trigonometric functions), while $f_{\mathrm{aug}}$ is supposed to capture the residual dynamics via flexible neural networks. When the symbolic structure is fixed, the two components can be provably separated via simple L2 regularization of $f_{\mathrm{aug}}$. This works, because the resulting optimization problem is convex and orthogonality $f_{\mathrm{aug}} \perp f_{\mathrm{phy}}$ is guaranteed by the properties of L2 projection.

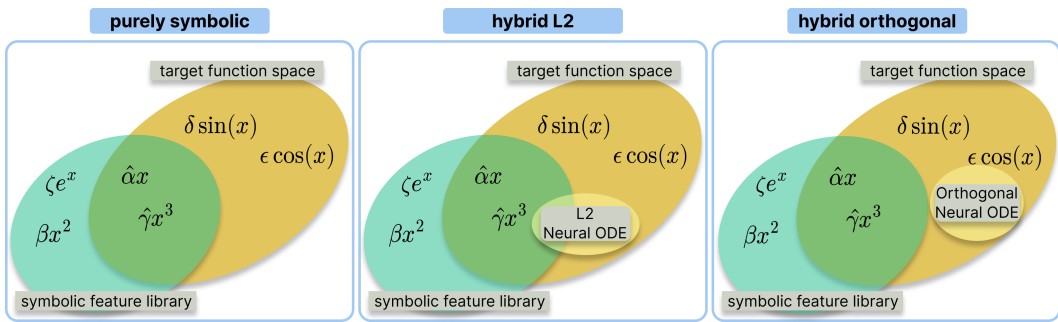

Figure 1: Symbolic and symbolic-neural models for the assumed true system $f = \alpha x + \gamma x^3 + \delta \sin(x) + \epsilon \cos(x)$. **Left:** A limited symbolic library could capture $\hat{f} = \hat{\alpha} x + \hat{\gamma} x^3$ resulting in both imperfect reconstruction and incorrect estimation of $\alpha$ and $\gamma$. **Middle:** A naive hybrid L2-regularized model could yield $\hat{f} = \hat{\alpha} x + \hat{\gamma} x^3 + f_{\mathrm{aug}, L_2}$, where the minimum L2 $f_{\mathrm{aug}, L_2}$ may still overlap with the symbolic feature library. It can achieve good trajectory recovery, but may still not consistently estimate $\alpha$ and $\gamma$. **Right:** Our OrthoReg model explicitly regularizes the neural component $f_{\mathrm{aug}, \mathrm{orth}}$ to be orthogonal to the feature library, resulting in $\hat{f} = \hat{\alpha} x + \hat{\gamma} x^3 + f_{\mathrm{aug}, \mathrm{orth}}$ that also properly estimates $\alpha$ and $\gamma$.

When also learning $f_{\mathrm{phy}}$ via symbolic regression from the same data, simply optimizing the residual component subject to an L2 constraint "$\min \|f_{\mathrm{aug}}\|_2$" does not guarantee orthogonality $f_{\mathrm{phy}} \perp f_{\mathrm{aug}}$ in the optimum of this non-convex problem. Hence, APHYNITY's approach cannot be transferred to this setting. Concretely, we consider learning the the symbolic part via a SINDy (Brunton et al., 2016) like approach: assume $f_{\mathrm{phy}}$ is some linear combination of (non-linear) basis function $\{\phi_j\}_{j=1}^k$ from some fixed, but potentially large library and fit the coefficients via sparse regression. For the residual neural component, we allow arbitrary neural networks essentially leading to a neural ODE (NODE) Chen et al. (2018). Figure 1 illustrates the fundamental challenge: **Left:** Most library-based pure symbolic regression approaches, especially the much celebrated SINDy (Brunton et al., 2016), are still limited by the size of the library. Hence, complex residual phenomena present in the target dynamics may still not lie within the linear span of the library functions—a hybrid approach is paramount. **Middle:** When naively extending L2 regularization-based approaches, like APHYNITY (Yin et al., 2021), to settings where also the symbolic component is learned, the neural component, despite small in "magnitude" (L2 norm), may still capture functions in $\mathcal{F}_{\mathrm{phy}}$. **Right:** OrthReg (ours) ensures that the neural component $f_{\mathrm{aug}}$ only captures aspects outside of $\mathcal{F}_{\mathrm{phy}}$.

In this work, we introduce a theoretically grounded and practically effective method to learn hybrid dynamical systems, where the mechanistic component is discovered from data via symbolic regression while ensuring that the residual neural component remains orthogonal to the symbolic part. Concretely, we provide

- **theoretical analysis** of OrthoReg as a consistent and efficient method to ensure $f_{\mathrm{aug}} \perp \mathcal{F}_{\mathrm{phy}}$.
- an **algorithmic solution** that accommodates symbolic regression with sparsity penalties while still explicitly enforcing orthogonality.
- thorough **empirical validation** of OrthoReg demonstrating improved out-of-distribution generalization and symbolic identification compared to existing methods.[1]

## 2 RELATED WORK

Methods for uncovering governing dynamical laws from data span a broad range, striking different balances between interpretability and expressiveness. We survey the main works that motivate our orthogonal regularization scheme.

---

[1]All code will be available at [anonymized].

**(Dynamic) symbolic regression.** Symbolic regression recovers interpretable mathematical expressions using genetic programming (Koza, 1994; Schmidt & Lipson, 2009), deep learning architectures (Petersen et al., 2019; 2021), or "sparse library" approaches like SINDy (Brunton et al., 2016). Recent advances incorporate physical constraints such as matching units (Tenachi et al., 2023) or employ large-scale pre-training to scale inference (Becker et al., 2023; d'Ascoli et al., 2024) enable large-scale generation. We focus on settings, where the true underlying dynamics consist of one part that can be composed from known library functions and another possibly complex non-linear part that cannot easily be captured exactly symbolically without hampering interpretability.

**Physics-informed neural networks.** PINNs (Raissi et al., 2019) embed differential equations as soft constraints for mesh-free solutions, while Universal ODEs (Rackauckas et al., 2020) parameterize unknown terms with neural networks. Comprehensive surveys (Cuomo et al., 2022; Hao et al., 2022) establish these as major paradigms for scientific machine learning, but both approaches rely on explicitly encoding prior knowledge of the governing physical laws, which limits flexibility when such knowledge is incomplete or uncertain.

**Neural, symbolic, and hybrid methods.** Hybrid approaches combine symbolic interpretability with neural flexibility. Rudy et al. (2017) pioneered combining PINNs with sparse regression for PDE discovery, while recent work extends frameworks to gray-box learning with symbolic regression coupled to extended PINNs (Chen et al., 2021; Kiyani et al., 2023). For ODE discovery, the APHYNITY framework (Yin et al., 2021) provides theoretical foundations for hybrid decompositions $f = f_{\text{phy}} + f_{\text{aug}}$ with existence and uniqueness guarantees, but critically assumes fixed symbolic structures and disallows simultaneous discovery of the symbolic and neural components.

Pure neural approaches for ODE learning have been extended to incorporate "soft knowledge" such as sparsity (Aliee et al., 2022), manifold/conservation constraints (Greydanus et al., 2019; Matsubara & Yaguchi, 2022; White et al., 2023), or meta-learning techniques for optimizing physics-ML trade-offs (Mouli et al., 2024). However, these either lack symbolic interpretability or, in the case of the latter, do not address overlap challenge in the symbolic-neural decomposition—again compromising interpretability of the symbolic part.

## 3 BACKGROUND

### 3.1 PROBLEM SETUP

Let $\mathcal{F}$ be a Hilbert space of functions $f : \mathbb{R}^n \to \mathbb{R}^n$. We will primarily consider $L^2$ spaces either with respect to the Lebesgue measure or an empirical measure given by a finite dataset $\mathcal{D}$. In the latter case, we write $\|\cdot\|_{\mathcal{D}}$ and $\langle\cdot,\cdot\rangle_{\mathcal{D}}$ for the norm and inner product on $\mathcal{F}$. The functions $f \in \mathcal{F}$ are interpreted as vector fields of autonomous, first order differential equations

$$\frac{\mathrm{d}x}{\mathrm{d}t} = f(x), \qquad \text{with solution trajectories} \, x : \mathbb{R} \to \mathbb{R}^n \,.$$

Following prior work (Yin et al., 2021; Rackauckas et al., 2020), we assume a decomposition

$$f = f_{\text{phy}} + f_{\text{aug}}, \qquad f_{\text{phy}} \in \mathcal{F}_{\text{phy}}, \, f_{\text{aug}} \in \mathcal{F}.$$

of vector fields of interest into a "physical" (or symbolic/mechanistic) component and an "augmented" (or neural/residual) component. The space $\mathcal{F}_{\text{phy}} \subseteq \mathcal{F}$ of candidate symbolic components is typically restricted to functions that can be represented in closed form using known functions to be amenable to direct interpretation and dissemination by humans.

Most existing methods assume $f_{\text{phy}}$ to be either known exactly, or to be given as a parametric family, where only a (usually small) set of parameters is unknown. Practically, this is often implemented via a linear combination of non-linear basis functions approach:

$$f_{\text{phy}} \in \mathcal{F}_{\text{phy}} = \left\{ \sum_{i=1}^{M} \alpha_i \phi_i \mid \alpha_i \in \mathbb{R} \right\} \text{ for fixed dictionary functions } \phi_i : \mathbb{R}^n \to \mathbb{R}^n \,. \quad (1)$$

The dynamics governing most real systems are not perfectly described by such simple closed-form expressions, but contain higher-order effects or complex interactions that are rarely captured by

simple interpretable mathematical expressions. To capture such residual effects the augmentation $f_{\mathrm{aug}} \in \mathcal{F}$ is supposed to be flexible and expressive, albeit potentially not easily interpretable. Hence, a natural choice to represent $f_{\mathrm{aug}}$ is via flexible function approximators such as neural networks, giving rise to the term "neural component." Crucially, the neural component should *only capture effects that cannot be captured by the symbolic component*.

In the current formulation, one could simply set $f_{\mathrm{aug}} \equiv f$ and $f_{\mathrm{phy}} \equiv 0$. However, this would undermine the entire idea of hybrid modeling. When $f_{\mathrm{phy}}$ is known, Yin et al. (2021) provide thorough theoretical guarantees showing that a relatively simple norm-based regularization scheme is sufficient to ensure that $f_{\mathrm{aug}}$ "only captures what is necessary, but not more." The corresponding optimization problem solved in practice is

$$\min_{f_{\mathrm{phy}} \in \mathcal{F}_{\mathrm{phy}}, f_{\mathrm{aug}} \in \mathcal{F}_{\mathrm{aug}}} \|f - f_{\mathrm{phy}} - f_{\mathrm{aug}}\|_{\mathcal{D}}^2 + \lambda \|f_{\mathrm{aug}}\|_{\mathcal{D}}^2 \,. \tag{2}$$

For a fixed $f_{\mathrm{phy}}$, the minimum of eq. (2) with respect to $f_{\mathrm{aug}}$ is given by

$$\hat{f}_{\mathrm{aug}} = \tfrac{1}{1+\lambda} \left(f - f_{\mathrm{phy}}\right),$$

so that eq. (2) reduces to the best-approximation problem

$$\min_{f_{\mathrm{phy}} \in \mathcal{F}_{\mathrm{phy}}} \|f - f_{\mathrm{phy}}\|_{\mathcal{D}}^2 \,.$$

If $\mathcal{F}_{\mathrm{phy}}$ is a closed linear subspace of $\mathcal{F}$, for example as in eq. (1), the Hilbert space projection theorem (Lax, 2002) ensures that the minimizer is the orthogonal projection $P_{\mathcal{F}_{\mathrm{phy}}}(f)$, and the residual $f - P_{\mathcal{F}_{\mathrm{phy}}}(f)$ (hence $\hat{f}_{\mathrm{aug}}$) is orthogonal to $\mathcal{F}_{\mathrm{phy}}$. Keeping the general intuition intact, APHYNITY proves existence and uniqueness of the projection as best approximation under more general geometric assumptions such as *proximinality* and *Chebyshevness* of $\mathcal{F}_{\mathrm{phy}}$ (Yin et al., 2021).

## 3.2 EXTENSION TO SPARSE SYMBOLIC DISCOVERY

A natural extension to a fully known $f_{\mathrm{phy}}$ or the structure being known up to a small set of parameters, is to allow for a sparse linear combination of a potentially large collection of non-linear dictionary functions like in SINDy (Brunton et al., 2016). After fixing the candidate basis functions $\{\phi_i\}_{i=1}^M$ we select only a small support set $S \subset \{1, \ldots, M\}$ of basis functions that enter the expression with non-zero coefficients. The induced function space is

$$\mathcal{F}_{\mathrm{phy}}(S) := \mathrm{span}\{\phi_j \mid j \in S\}.$$

In practice, the set $S$ is fitted via sparse regression methods (e.g., L1 regularization $\|\cdot\|_1$ or more involved iterated sparse regressions as in SINDy) to encourage small supports $S$.

While at first this appears to be a natural extension to APHYNITY, at closer inspection this breaks the assumptions required for APHYNITY's guarantees. When $\mathcal{F}_{\mathrm{phy}}$ itself is learned together with the support $S$, the optimization problem becomes combinatorial and non-convex such that projection theory no longer applies, and the augmentation can "re-learn" components of the symbolic space, see fig. 1. In this setting, $L^2$ regularization, while controlling the *magnitude*, but not the *direction* of $f_{\mathrm{aug}}$ relative to $\mathcal{F}_{\mathrm{phy}}(S)$.

This is the fundamental gap our work addresses: expressive (sparse) symbolic discovery requires additional techniques to ensure that neural augmentations do not overlap with the symbolic component. A complete analysis is given in appendix A.

## 3.3 EMPIRICAL ORTHOGONALITY CONSTRAINTS

Consider a dataset of observations $\mathcal{D} = \{x_i\}_{i=1}^N \subset \mathbb{R}^n$ that define the empirical ($L^2$) inner product

$$\langle \cdot, \cdot \rangle_{\mathcal{D}} : \mathcal{F} \times \mathcal{F} \to \mathbb{R}, \ \langle f, g \rangle_{\mathcal{D}} = \frac{1}{N} \sum_{i=1}^N f(x_i)^\top g(x_i) \,. \tag{3}$$

The OrthoReg regularizer then directly enforces orthogonality between $f_{\mathrm{aug}}$ and $\mathcal{F}_{\mathrm{phy}}$ with respect to this empirical inner product via

$$\langle f_{\mathrm{aug}}, \phi_j \rangle_{\mathcal{D}} \overset{!}{=} 0, \quad \text{for all } j \in S \,,$$

ensuring that augmentations only capture functions outside the capacity of the symbolic functions. All details are provided in appendix B.

## 4 METHOD: ORTHOREG FOR HYBRID MODELING

### 4.1 EXPLICIT ORTHOGONALITY CONSTRAINTS

Instead of relying on implicit orthogonality from $L^2$ regularization, we enforce it explicitly. Given basis functions $\{\phi_j\}_{j=1}^M$ spanning $\mathcal{F}_{\text{phy}}$ and neural augmentation $\hat{f}_{\text{aug}}$, our overall orthogonality penalty reads

$$\mathcal{L}_{\text{reg}}^{\perp} = \lambda \sum_{j=1}^{k} \left\langle \hat{f}_{\text{aug}}, \phi_j \right\rangle_{\mathcal{D}}^2, \tag{4}$$

where $\lambda \in \mathbb{R}_{\geq 0}$ is a regularization parameter.

**Theorem 4.1** (Orthogonality at Optimum [informal]). *The orthogonality penalty $\mathcal{L}_{\text{reg}}^{\perp}$ ensures that at the global minimum, $\hat{f}_{\text{aug}} \perp \mathcal{F}_{\text{phy}}$ with respect to the empirical inner product.*

*Proof idea.* Quadratic penalty theory (Bertsekas, 1976; 1999) and the analysis in appendix B.4 show that increasing $\lambda$ enforces $\hat{f}_{\text{aug}} \perp \mathcal{F}_{\text{phy}}$ at stationary points of the penalized loss. $\square$

### 4.2 THEORETICAL GUARANTEES

Our theoretical analysis establishes the key distinction between OrthoReg and L2 regularization approaches, providing formal guarantees for orthogonal hybrid modeling.

**Orthogonality Enforcement** Standard quadratic penalty theory (Bertsekas, 1976; 1999) ensures that increasing $\lambda$ forces optimization algorithms to satisfy the orthogonality constraints in the limit, with stationary points approaching exact orthogonality under standard SGD convergence assumptions (Ghadimi & Lan, 2013).

**Approximation Quality** Under orthogonality constraints, our hybrid model satisfies

$$\|f - \hat{f}\|_{\mathcal{D}} \leq \|f - P_{\mathcal{F}_{\text{phy}}}^{\mathcal{D}}(f)\|_{\mathcal{D}} + \epsilon_{\text{neural}}(\lambda), \tag{5}$$

where the first term represents the irreducible approximation error from symbolic library limitations, and $\epsilon_{\text{neural}}(\lambda)$ represents the neural network approximation error in the orthogonal complement space, with $\epsilon_{\text{neural}}(\lambda) \to 0$ as orthogonality strength increases and neural network capacity grows.

**L2 vs. Orthogonal Regularization** The fundamental distinction lies in constraint specificity. L2 regularization controls magnitude through the decomposition

$$\|\hat{f}_{\text{aug}}\|_{\mathcal{D}}^2 = \sum_j \langle \hat{f}_{\text{aug}}, \phi_j \rangle_{\mathcal{D}}^2 + \|\hat{f}_{\text{aug}} - P_{\mathcal{F}_{\text{phy}}}^{\mathcal{D}}(\hat{f}_{\text{aug}})\|_{\mathcal{D}}^2$$

where the equality follows from the orthogonal decomposition and Pythagorean theorem in inner product spaces (Rudin, 1987). Even when this total is small, individual inner products $\langle \hat{f}_{\text{aug}}, \phi_j \rangle_{\mathcal{D}}$ can be non-zero, allowing neural-symbolic overlap. When $\mathcal{F}_{\text{phy}}$ is learned through sparsity constraints, the resulting non-convex optimization landscape exacerbates this issue, which orthogonality constraints explicitly prevent.

**Finite-Sample Guarantees** For bounded functions with $|\hat{f}_{\text{aug}}(x_i)^\top \phi_j(x_i)| \leq M$ and training set size $N$, empirical orthogonality $\langle \hat{f}_{\text{aug}}, \phi_j \rangle_{\mathcal{D}} = 0$ provides finite-sample control over the population inner product. By Hoeffding's inequality (Hoeffding, 1963), the population inner product satisfies

$$|\mathbb{E}[\hat{f}_{\text{aug}}(X)^\top \phi_j(X)]| = O(M/\sqrt{N})$$

with high probability, providing concrete bounds on how well the orthogonal decomposition generalizes beyond the training set under these boundedness assumptions.

This theoretical foundation ensures that OrthoReg creates truly complementary representations where symbolic components capture all dynamics within their span, while neural components model only residual dynamics. Complete proofs and additional theoretical analysis are provided in appendix C.

---

**Algorithm 1** OrthoReg Training

---

1: **Input:** Data $(x_i, y_i)$, basis functions $\{\phi_j\}$, regularization weight $\lambda$, sparsity weight $\mu$
2: Initialize symbolic coefficients $w$, neural parameters $\theta$
3: **for** each epoch **do**
4:    Forward: $\hat{f} = \sum_j w_j \phi_j(x_i) + \hat{f}_{\text{aug}}(x_i; \theta)$
5:    Compute fit loss: $L_{\text{fit}} = \|y_i - \hat{f}\|^2$
6:    Compute orthogonality penalty: $L_{\text{orth}} = \lambda \sum_j \langle \hat{f}_{\text{aug}}, \phi_j \rangle_{\mathcal{D}}^2$
7:    Compute sparsity penalty: $L_{\text{sparse}} = \mu \|w\|_1$
8:    Update $\theta, w$ via $\nabla(L_{\text{fit}} + L_{\text{orth}} + L_{\text{sparse}})$
9: **end for**

---

## 4.3 Monte Carlo Approximation

In practice, the orthogonality penalty requires Monte Carlo approximation over minibatches:

$$\widehat{\mathcal{L}}_{\text{reg}}^{\perp} = \lambda \sum_{j=1}^{k} \left( \frac{1}{B} \sum_{i=1}^{B} \hat{f}_{\text{aug}}(x_i)^{\top} \phi_j(x_i) \right)^2 \tag{6}$$

The batch approximation error scales as $O(1/\sqrt{B})$ with high probability, ensuring convergence while maintaining computational efficiency. This stochastic approximation provides implicit regularization benefits during training. Detailed analysis of batch approximation quality, convergence rates, and practical implications are provided in appendix D. An ablation on the number of samples is shown in appendix E.

## 4.4 Implementation and Computational Considerations

Our implementation works with $k$ basis functions $\{\phi_j\}_{j=1}^{k}$ in the symbolic library $\mathcal{F}_{\text{phy}}$, input dimension $d$, and sparsity regularization strength $\mu$. algorithm 1 sketches the OrthoReg training procedure. The orthogonality computation requires $\mathcal{O}(kBd)$ operations per forward pass with modest 5-15% computational overhead. OrthoReg integrates with sparsity constraints:

$$\min_{w,\theta} \|f - (\hat{f}_{\text{phy}} + \hat{f}_{\text{aug}})\|^2 + \mu \|w\|_1 + \lambda \sum_j \langle \hat{f}_{\text{aug}}, \phi_j \rangle_{\mathcal{D}}^2 \tag{7}$$

# 5 Experiments

We evaluate OrthoReg across three dynamical systems of increasing complexity: a modified damped pendulum, a Lotka–Volterra predator-prey system, and a memory-modulated SIR epidemiological model. Our evaluation focuses on three complementary metrics: (i) trajectory accuracy measured by normalized mean-squared error (MSE) on derivatives and integrated states[2], (ii) symbolic recovery quality measured by F1 score, and (iii) component separation quantified via an orthogonality measure[3]. We compare three hybrid modeling variants: pure symbolic regression (SINDy), L2-regularized, and OrthoReg-regularized hybrid models. Each experiment is repeated over five stochastic runs to ensure robust conclusions.

## 5.1 Damped Pendulum: Missing Dynamics

The modified damped pendulum system exhibits dynamics similar to the classical driven damped pendulum (Kharkongor & Mahato, 2018) and include higher-order nonlinear terms absent from the feature library:

$$\ddot{\theta} + \alpha\dot{\theta} + \sin(\theta) + \beta_1\theta^3 + \beta_2\dot{\theta}^3 + \beta_3\sin(3\theta) = 0, \tag{8}$$

---

[2]MSE values are normalized by the squared norm of the target signal for scale invariance.

[3]Orthogonality $= \frac{1}{k} \sum_{j=1}^{k} \frac{|\langle \hat{f}_{\text{aug}}, \phi_j \rangle_{\mathcal{D}}|}{\|\hat{f}_{\text{aug}}\|_{\mathcal{D}} \|\phi_j\|_{\mathcal{D}}}$

Table 1: Performance in the medium missing dynamics regime. OrthoReg achieves superior predictive accuracy and symbolic identification across all metrics.

| Metric | Pure | L2 | OrthoReg |
|---|---|---|---|
| **In-Distribution Performance** | | | |
| ID Deriv MSE ($\downarrow$) | $6.9{\times}10^{-2} \pm 7.0{\times}10^{-6}$ | $6.9{\times}10^{-2} \pm 4.0{\times}10^{-6}$ | $\mathbf{1.4{\times}10^{-2} \pm 7.9{\times}10^{-5}}$ |
| ID State MSE ($\downarrow$) | $4.9{\times}10^{-2} \pm 1.2{\times}10^{-3}$ | $5.3{\times}10^{-2} \pm 1.2{\times}10^{-3}$ | $\mathbf{1.1{\times}10^{-2} \pm 1.1{\times}10^{-3}}$ |
| ID Extra Deriv MSE ($\downarrow$) | $6.1{\times}10^{0} \pm 2.5{\times}10^{0}$ | $6.2{\times}10^{0} \pm 2.5{\times}10^{0}$ | $\mathbf{3.3{\times}10^{0} \pm 2.5{\times}10^{0}}$ |
| **Out-of-Distribution Performance** | | | |
| OOD T2 Deriv MSE ($\downarrow$) | $1.1{\times}10^{-1} \pm 1.0{\times}10^{-4}$ | $1.1{\times}10^{-1} \pm 4.9{\times}10^{-5}$ | $\mathbf{4.5{\times}10^{-2} \pm 7.3{\times}10^{-4}}$ |
| OOD T3 Deriv MSE ($\downarrow$) | $6.8{\times}10^{0} \pm 2.2{\times}10^{-1}$ | $6.9{\times}10^{0} \pm 1.0{\times}10^{-1}$ | $\mathbf{6.8{\times}10^{-1} \pm 1.3{\times}10^{-1}}$ |
| **System Identification Quality** | | | |
| F1 Score ($\uparrow$) | $4.7{\times}10^{-1} \pm 3.0{\times}10^{-2}$ | $4.7{\times}10^{-1} \pm 2.0{\times}10^{-2}$ | $\mathbf{9.3{\times}10^{-1} \pm 1.5{\times}10^{-1}}$ |
| Nonzero Terms ($\downarrow$) | $9.8{\times}10^{0} \pm 8.0{\times}10^{-1}$ | $9.8{\times}10^{0} \pm 4.0{\times}10^{-1}$ | $\mathbf{3.6{\times}10^{0} \pm 1.3{\times}10^{0}}$ |
| Orthogonality ($\uparrow$) | – | $1.4{\times}10^{-1} \pm 1.3{\times}10^{-1}$ | $\mathbf{2.8{\times}10^{-1} \pm 2.0{\times}10^{-1}}$ |

where $\beta$ terms represent effects absent from the symbolic library. Five stochastic runs are used to ensure robust conclusions.

Table 1 shows performance under medium-missing dynamics (mean $\beta = 0.6$ for $\beta_i$ in eq. (8)). Derivative and state MSE quantify trajectory fit, the F1 score measures symbolic recovery against ground-truth terms, and the orthogonality score reflects separation between symbolic and neural components. Under these metrics, OrthoReg reduces in-distribution derivative MSE from $6.9 \cdot 10^{-2}$ (Pure/L2) to $1.4 \cdot 10^{-2}$, out-of-distribution derivative error under initial condition perturbation (OOD T2 drops from $\sim 0.11$ to $0.045$) and under parameter perturbation (OOD T3) from $\sim 6.8$ to $0.68$, and symbolic recovery improves from F1 $0.47$ to $0.93$. OrthoReg produces fewer redundant terms ($3.6$ vs $9.8$) and higher orthogonality ($0.28$ vs $0.14$), indicating effective separation of complementary components. These results suggest that the orthogonality prior does not simply improve fit: it encourages complementary component representations that transfer beyond the training distribution.

## 5.2 Cross-System Validation

To test robustness across systems and complexity, we evaluate OrthoReg on a Lotka–Volterra predator-prey system and a memory-modulated SIR model (appendix F). The Lotka–Volterra system introduces coupled temporal dynamics, while the SIR model adds state-dependent time scales and memory effects. OrthoReg shows modest gains in Lotka–Volterra (3–5% OOD improvement, F1 $0.24$ vs $0.22$) and maintains strong orthogonality in the challenging SIR model ($0.80$ vs $0.17$ for L2), though all approaches struggle with symbolic recovery in this complex system. OrthoReg achieves superior sparsity ($9.6$ vs $44.0$ terms for pure symbolic), demonstrating that orthogonal regularization effectively enforces component separation even when symbolic discovery is difficult.

## 5.3 Comparison with Pure Neural Baselines

Table 2: Baseline comparison in medium missing dynamics regime ($\beta = 0.6$). OrthoReg uniquely provides symbolic recovery while achieving competitive predictive performance.

| Metric | PINN | Universal ODE | OrthoReg |
|---|---|---|---|
| ID Deriv MSE ($\downarrow$) | $8.63{\times}10^{-2} \pm 4.38{\times}10^{-4}$ | $\mathbf{4.81{\times}10^{-3} \pm 4.30{\times}10^{-4}}$ | $1.40{\times}10^{-2} \pm 7.90{\times}10^{-5}$ |
| OOD T2 Deriv MSE ($\downarrow$) | $2.10{\times}10^{-1} \pm 2.00{\times}10^{-2}$ | $1.30{\times}10^{-1} \pm 4.00{\times}10^{-2}$ | $\mathbf{4.50{\times}10^{-2} \pm 7.30{\times}10^{-4}}$ |
| OOD T3 Deriv MSE ($\downarrow$) | $2.00{\times}10^{0} \pm 6.00{\times}10^{-2}$ | $\mathbf{4.00{\times}10^{-1} \pm 6.00{\times}10^{-2}}$ | $6.80{\times}10^{-1} \pm 1.30{\times}10^{-1}$ |
| F1 Score ($\uparrow$) | – | – | $\mathbf{9.3{\times}10^{-1} \pm 1.5{\times}10^{-1}}$ |
| Orthogonality ($\uparrow$) | – | – | $\mathbf{2.8{\times}10^{-1} \pm 2.0{\times}10^{-1}}$ |

We also compare to pure neural approaches in table 2, including PINNs (Raissi et al., 2019) and Universal Differential Equations (Rackauckas et al., 2020). While these baselines achieve competitive trajectory fitting, they cannot recover symbolic components. In contrast, OrthoReg matches predictive performance while providing interpretable representations, demonstrating the benefit of hybrid modeling for scientific discovery. Implementation details are in appendix G.

## 5.4 DATASET DIFFICULTY ABLATION

Table 3: Dataset difficulty ablation across missing dynamics regimes indicated by mean effect strength absent from the symbolic library. OrthoReg consistently improves OOD predictive performance and symbolic recovery, while ID performance remains strong across all methods.

| Difficulty | Metric | Pure | L2 | OrthoReg |
|---|---|---|---|---|
| Low $(\beta = 0.077)$ | ID Deriv MSE ($\downarrow$) | $\mathbf{7.8 \times 10^{-4} \pm 1.5 \times 10^{-4}}$ | $1.1 \times 10^{-3} \pm 0.1 \times 10^{-3}$ | $2.7 \times 10^{-3} \pm 0.4 \times 10^{-3}$ |
| | OOD T2 MSE ($\downarrow$) | $\mathbf{1.5 \times 10^{-3} \pm 0.2 \times 10^{-3}}$ | $2.0 \times 10^{-3} \pm 0.0 \times 10^{-3}$ | $2.8 \times 10^{-3} \pm 0.1 \times 10^{-3}$ |
| | OOD T3 MSE ($\downarrow$) | $1.6 \times 10^{2} \pm 1.5 \times 10^{2}$ | $5.9 \pm 0.0$ | $\mathbf{5.8 \times 10^{-1} \pm 0.5 \times 10^{-1}}$ |
| | F1 Score ($\uparrow$) | $5.0 \times 10^{-1} \pm 0.8 \times 10^{-1}$ | $7.2 \times 10^{-1} \pm 0.4 \times 10^{-1}$ | $\mathbf{8.6 \times 10^{-1} \pm 0.0 \times 10^{-1}}$ |
| | Orthogonality ($\uparrow$) | – | $2.6 \times 10^{-2} \pm 2.3 \times 10^{-2}$ | $\mathbf{6.2 \times 10^{-1} \pm 1.9 \times 10^{-1}}$ |
| Medium $(\beta = 0.6)$ | ID Deriv MSE ($\downarrow$) | $6.9 \times 10^{-2} \pm 0.0 \times 10^{-2}$ | $6.9 \times 10^{-2} \pm 0.0 \times 10^{-2}$ | $\mathbf{1.4 \times 10^{-2} \pm 0.0 \times 10^{-2}}$ |
| | OOD T2 MSE ($\downarrow$) | $9.3 \times 10^{-2} \pm 0.0 \times 10^{-2}$ | $9.3 \times 10^{-2} \pm 0.0 \times 10^{-2}$ | $\mathbf{1.5 \times 10^{-2} \pm 0.0 \times 10^{-2}}$ |
| | OOD T3 MSE ($\downarrow$) | $6.8 \pm 0.2$ | $6.9 \pm 0.1$ | $\mathbf{6.8 \times 10^{-1} \pm 1.2 \times 10^{-1}}$ |
| | F1 Score ($\uparrow$) | $4.7 \times 10^{-1} \pm 0.3 \times 10^{-1}$ | $4.7 \times 10^{-1} \pm 0.2 \times 10^{-1}$ | $\mathbf{9.3 \times 10^{-1} \pm 1.3 \times 10^{-1}}$ |
| | Orthogonality ($\uparrow$) | – | $1.4 \times 10^{-1} \pm 1.2 \times 10^{-1}$ | $\mathbf{2.8 \times 10^{-1} \pm 1.8 \times 10^{-1}}$ |
| High $(\beta = 2.0)$ | ID Deriv MSE ($\downarrow$) | $3.9 \times 10^{-2} \pm 0.1 \times 10^{-2}$ | $3.9 \times 10^{-2} \pm 0.1 \times 10^{-2}$ | $\mathbf{3.9 \times 10^{-2} \pm 0.1 \times 10^{-2}}$ |
| | OOD T2 MSE ($\downarrow$) | $4.5 \times 10^{-2} \pm 0.2 \times 10^{-2}$ | $4.5 \times 10^{-2} \pm 0.1 \times 10^{-2}$ | $\mathbf{4.4 \times 10^{-2} \pm 0.1 \times 10^{-2}}$ |
| | OOD T3 MSE ($\downarrow$) | $4.5 \times 10^{-1} \pm 2.3 \times 10^{-1}$ | $4.1 \times 10^{-1} \pm 0.7 \times 10^{-1}$ | $\mathbf{2.6 \times 10^{-1} \pm 0.4 \times 10^{-1}}$ |
| | F1 Score ($\uparrow$) | $4.6 \times 10^{-1} \pm 0.5 \times 10^{-1}$ | $4.3 \times 10^{-1} \pm 0.4 \times 10^{-1}$ | $\mathbf{5.2 \times 10^{-1} \pm 0.6 \times 10^{-1}}$ |
| | Orthogonality ($\uparrow$) | – | $\mathbf{7.9 \times 10^{-1} \pm 0.7 \times 10^{-1}}$ | $4.3 \times 10^{-1} \pm 1.4 \times 10^{-1}$ |

The effectiveness of OrthoReg depends on how much the system exceeds the symbolic library. In low-missing regimes (mean $\beta = 0.077$ for $\beta_i$ in eq. (8)), all models perform comparably. In medium-missing regimes ($\beta = 0.6$), OrthoReg dramatically improves symbolic F1 (0.93 vs 0.47) and OOD T3 derivative error (0.68 vs 6.8), while in high-missing regimes ($\beta = 2.0$), symbolic recovery deteriorates across all methods, though OrthoReg still maintains a modest advantage. This ablation suggests that orthogonal regularization is most effective when the system partially exceeds the library, guiding complementary learning without overfitting trivial or impossible dynamics.

## 5.5 REGULARIZATION STRENGTH ABLATION

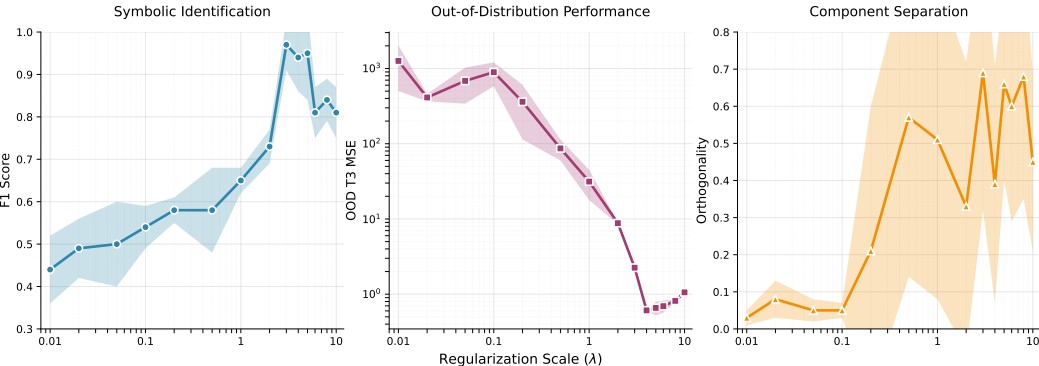

Figure 2: Regularization strength ablation. The optimal range is $\lambda \in [3.0, 5.0]$, achieving F1 scores above 0.95 with excellent OOD performance. Lower regularization leads to poor symbolic identification, while higher regularization maintains good performance but may over-constrain the model.

We investigate how the orthogonality regularization scale $\lambda$ affects OrthoReg (fig. 2). Too weak regularization fails to enforce complementary components, degrading symbolic identification and extrapolation, while overly strong regularization slightly constrains the model without harming predictions, leaving an optimal range for $\lambda$. Due to the correlation of trajectory and symbolic metrics, when applying OrthoReg to unknown systems we recommend scaling $\lambda$ relative to the base weights and monitoring symbolic F1 (if available) or orthogonality as a proxy to ensure complementary component formation.

## 5.6 Sampling Scheme Ablation

Table 4: Sampling scheme ablation (regular vs. irregular). OrthoReg maintains OOD predictive accuracy, symbolic recovery, and interpretability, even under irregular sampling.

| Sampling | Metric | Pure | L2 | OrthoReg |
|---|---|---|---|---|
| Regular | ID Deriv MSE ($\downarrow$) | $6.9\times10^{-2} \pm 7.0\times10^{-6}$ | $6.9\times10^{-2} \pm 4.0\times10^{-6}$ | $\mathbf{1.4\times10^{-2} \pm 7.9\times10^{-5}}$ |
| | OOD T2 MSE ($\downarrow$) | $0.11 \pm 1.0\times10^{-4}$ | $0.11 \pm 4.9\times10^{-5}$ | $\mathbf{0.045 \pm 7.3\times10^{-4}}$ |
| | OOD T3 MSE ($\downarrow$) | $6.8 \pm 0.22$ | $6.9 \pm 0.10$ | $\mathbf{0.68 \pm 0.13}$ |
| | F1 Score ($\uparrow$) | $0.47 \pm 0.03$ | $0.47 \pm 0.02$ | $\mathbf{0.93 \pm 0.15}$ |
| | Orthogonality ($\uparrow$) | $0.00 \pm 0.00$ | $0.14 \pm 0.13$ | $\mathbf{0.28 \pm 0.20}$ |
| Irregular | ID Deriv MSE ($\downarrow$) | $3.5 \pm 1.9\times10^{-4}$ | $3.5 \pm 2.1\times10^{-4}$ | $\mathbf{3.5 \pm 1.0\times10^{-4}}$ |
| | OOD T2 MSE ($\downarrow$) | $3.8 \pm 2.1\times10^{-3}$ | $3.8 \pm 2.5\times10^{-3}$ | $\mathbf{3.8 \pm 2.0\times10^{-3}}$ |
| | OOD T3 MSE ($\downarrow$) | $40.0 \pm 0.93$ | $39.0 \pm 1.0$ | $\mathbf{37.0 \pm 0.46}$ |
| | F1 Score ($\uparrow$) | $0.30 \pm 0.01$ | $0.30 \pm 0.01$ | $\mathbf{0.31 \pm 0.01}$ |
| | Orthogonality ($\uparrow$) | $0.00 \pm 0.00$ | $0.14 \pm 0.14$ | $\mathbf{0.37 \pm 0.28}$ |

We further test regular (uniform) versus irregular (non-uniform) time sampling. Irregular sampling reduces absolute performance across all methods, increasing derivative errors and lowering F1 scores. Nevertheless, OrthoReg retains relative advantages, including higher orthogonality (0.37 vs 0.14) and fewer redundant terms, demonstrating that orthogonal regularization benefits persist under realistic, non-ideal observation schemes.

Table 4 evaluates regular (uniform) versus irregular (non-uniform) sampling. Irregular sampling degrades absolute performance across all methods, yet OrthoReg retains relative advantages, including higher orthogonality (0.37 vs 0.14) and fewer nonzero terms (16.2 vs 16.8 and 17.2), demonstrating that orthogonal regularization benefits persist under realistic, non-ideal observation schemes. This demonstrates that orthogonal regularization benefits persist beyond idealized observation schemes, enhancing robustness in realistic data collection scenarios.

## 5.7 Summary

OrthoReg consistently improves hybrid modeling. It achieves substantially higher symbolic recovery (F1 0.93 vs 0.47 for L2) while maintaining superior out-of-distribution generalization. Orthogonal regularization effectively separates complementary components, and its benefits persist under irregular sampling and varying dataset difficulty. These results demonstrate that OrthoReg guides hybrid models to learn interpretable and transferable representations even when the symbolic library is partially misspecified.

# 6 Conclusion

Hybrid modeling promises the interpretability of symbolic structure with the flexibility of neural augmentation, as exemplified by APHYNITY. Yet, extending from fixed symbolic libraries to symbolic regression introduces sparsity constraints, making the optimization non-convex and breaking APHYNITY's guarantees. In this regime, L2 regularization controls only magnitude, not direction, allowing symbolic and neural terms to overlap.

We resolve this with **OrthoReg**, which enforces explicit orthogonality $\hat{f}\mathrm{aug} \perp \mathcal{F}\mathrm{phy}$ regardless of convexity. Our contributions span theoretical analysis of L2's failure, a principled algorithmic solution, and empirical validation showing improved generalization, symbolic recovery, and interpretability.

Limitations are discussed in Appendix I, with promising directions including extensions to non-gradient symbolic regression (e.g., PySINDy). More broadly, OrthoReg enables complementary representations where symbolic terms capture all recoverable dynamics and neural components model only residuals, paving the way for hybrid modeling as a practical tool in scientific domains ranging from biology to climate science.

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

# A WHEN L2 REGULARIZATION FAILS: RIGOROUS ANALYSIS

## A.1 APHYNITY'S PROBLEM FORMULATION

Following Yin et al. (2021), we adopt their problem formulation. APHYNITY seeks to decompose unknown dynamics $f$ as:

$$\hat{f} = \hat{f}_{\text{phy}} + \hat{f}_{\text{aug}}$$

where $\hat{f}_{\text{phy}} \in \mathcal{F}_{\text{phy}} = \text{span}\{\phi_j\}_{j=1}^k$ and $\hat{f}_{\text{aug}}$ is learned via neural networks. The key insight is that we estimate decompositions $\hat{f}_{\text{phy}}, \hat{f}_{\text{aug}}$ that may not perfectly reconstruct $f$.

APHYNITY's optimization problem is:

$$\min_{\hat{f}_{\text{phy}} \in \mathcal{F}_{\text{phy}}, \hat{f}_{\text{aug}}} \|f - \hat{f}_{\text{phy}} - \hat{f}_{\text{aug}}\|^2 + \lambda \|\hat{f}_{\text{aug}}\|^2 \tag{9}$$

This is the formulation from the APHYNITY paper, where we learn estimates that approximate the true dynamics while regularizing the augmentation magnitude.

## A.2 CONVEX VS. NON-CONVEX SETTINGS

**Proposition A.1** (APHYNITY's Convex Guarantee). *When $\mathcal{F}_{\text{phy}} = span\{\phi_j\}_{j=1}^k$ is a linear subspace and the optimization in equation 9 is convex, the minimizer satisfies $\hat{f}_{\text{aug}} \perp \mathcal{F}_{\text{phy}}$.*

*Proof.* Following the analysis by Yin et al. (2021), for fixed $\hat{f}_{\text{phy}}$, the optimal $\hat{f}_{\text{aug}}$ is:

$$\hat{f}_{\text{aug}} = \frac{1}{1+\lambda}(f - \hat{f}_{\text{phy}})$$

Substituting back, the problem reduces to:

$$\min_{\hat{f}_{\text{phy}} \in \mathcal{F}_{\text{phy}}} \frac{\lambda}{1+\lambda} \|f - \hat{f}_{\text{phy}}\|^2$$

When $\mathcal{F}_{\text{phy}} = \text{span}\{\phi_j\}_{j=1}^k$ is a linear subspace, this is the orthogonal projection problem: $\hat{f}_{\text{phy}} = P_{\mathcal{F}_{\text{phy}}}(f)$. By the projection theorem, the residual $f - P_{\mathcal{F}_{\text{phy}}}(f)$ is orthogonal to $\mathcal{F}_{\text{phy}}$, and thus $\hat{f}_{\text{aug}} \perp \mathcal{F}_{\text{phy}}$. $\square$

**Theorem A.2** (L2 Failure with Sparse Symbolic Regression). *When symbolic regression uses sparsity constraints (e.g., L1 penalties), creating non-convex optimization landscapes, L2 regularization alone does not guarantee $\hat{f}_{\text{aug}} \perp \mathcal{F}_{\text{phy}}$.*

*Proof.* With sparsity constraints, the optimization becomes:

$$\min_{\hat{f}_{\text{phy}} \in \mathcal{F}_{\text{phy}}, \hat{f}_{\text{aug}}} \|f - \hat{f}_{\text{phy}} - \hat{f}_{\text{aug}}\|^2 + \lambda \|\hat{f}_{\text{aug}}\|^2 + \mu \|w\|_1$$

where $w$ are the coefficients of $\hat{f}_{\text{phy}} = \sum_j w_j \phi_j(x)$.

The L1 penalty creates a non-convex optimization landscape where the learned $\hat{f}_{\text{phy}}$ may correspond to different sparse subsets of basis functions. Unlike the convex case, $\hat{f}_{\text{phy}}$ need not be the orthogonal projection onto the full span $\mathcal{F}_{\text{phy}} = \text{span}\{\phi_j\}_{j=1}^k$.

Therefore, $\hat{f}_{\text{aug}} = \frac{1}{1+\lambda}(f - \hat{f}_{\text{phy}})$ is not guaranteed to be orthogonal to $\mathcal{F}_{\text{phy}}$, since $\hat{f}_{\text{phy}}$ may only span a sparse subset of the full symbolic space. $\square$

### A.3 IMPLICATIONS FOR HYBRID MODELING

The above results demonstrate that L2 regularization is insufficient in non-convex settings. Even if $\|f_{\text{aug}}\|$ is small, $f_{\text{aug}}$ may not be orthogonal, leading to:

- **Interpretability loss:** neural components re-learn symbolic dynamics.
- **Identifiability failure:** multiple $(f_{\text{phy}}, f_{\text{aug}})$ pairs explain the data equally well.

This motivates explicit orthogonality constraints, which we introduce in the main text, to enforce separation regardless of convexity.

## B EMPIRICAL FUNCTION SPACES AND ORTHOGONALITY

Following APHYNITY's setup (Yin et al., 2021), we restrict attention to finite-dimensional subspaces $\mathcal{F}_{\text{phy}} = \text{span}\{\phi_j\}_{j=1}^k$, which is sufficient for our symbolic regression setting. More general nonlinear families require different projection arguments and are beyond our scope.

### B.1 PARAMETERIZED FUNCTION FAMILIES

We work with parameterized function families where functions are uniquely determined by their parameters. This approach ensures computational tractability while maintaining theoretical rigor.

**Definition B.1** (Parameterized Function Family). *Let $\mathcal{X} \subset \mathbb{R}^n$ be a state space, and let $\mathcal{D} = \{x_i\}_{i=1}^N$ be a dataset drawn from distribution $\mu$. We work with functions $f : \mathcal{X} \to \mathbb{R}^d$ from parameterized families where functions are uniquely determined by their parameters. For computational purposes, we evaluate these functions only on the dataset $\mathcal{D}$.*

### B.2 EMPIRICAL INNER PRODUCT AND NORM

**Definition B.2** (Empirical Inner Product). *The empirical inner product on parameterized functions evaluated on $\mathcal{D}$ is defined as: $\langle f, g \rangle_{\mathcal{D}} = \frac{1}{N} \sum_{i=1}^N f(x_i)^\top g(x_i)$.*

*This induces the empirical norm: $\|f\|_{\mathcal{D}} = \sqrt{\langle f, f \rangle_{\mathcal{D}}}$.*

The empirical inner product endows the space $\mathcal{F}_{\mathcal{D}} = \{(f(x_1), \ldots, f(x_N)) : f : \mathcal{X} \to \mathbb{R}^d\}$ with the structure of a finite-dimensional inner product space (and hence a Hilbert space).

**Why This Matters:** The empirical inner product is:

- **Computable:** Can be evaluated on finite data
- **Theoretically Sound:** Provides an inner product structure in finite dimensions
- **Practically Relevant:** Directly corresponds to our implementation

### B.3 ORTHOGONALITY IN EMPIRICAL SPACES

**Definition B.3** (Empirical Orthogonality). *Two functions $f, g \in \mathcal{F}_{\mathcal{D}}$ are empirically orthogonal if $\langle f, g \rangle_{\mathcal{D}} = 0$.*

### B.4 EMPIRICAL PROJECTION THEOREM

**Theorem B.4** (Empirical Projection Theorem). *Let $\mathcal{F}_{\text{phy}} = \text{span}\{\phi_j\}_{j=1}^k$ be a finite-dimensional subspace of $\mathcal{F}_{\mathcal{D}}$, and let $f \in \mathcal{F}_{\mathcal{D}}$. Assume that $\{\phi_j\}_{j=1}^k$ are linearly independent on $\mathcal{D}$, i.e. no nontrivial linear combination vanishes simultaneously at all $x_i \in \mathcal{D}$. Then there exists a unique orthogonal decomposition: $f = f_{\text{phy}} + r$, where $f_{\text{phy}} \in \mathcal{F}_{\text{phy}}$ and $r \perp \mathcal{F}_{\text{phy}}$ with respect to the empirical inner product.*

*Proof.* We show both existence and uniqueness.

**Existence:** Let $\{\phi_j\}_{j=1}^k$ be a basis for $\mathcal{F}_{\text{phy}}$. We seek coefficients $\{w_j\}_{j=1}^k$ such that $f_{\text{phy}} = \sum_{j=1}^k w_j \phi_j$ and $r = f - f_{\text{phy}}$ is orthogonal to $\mathcal{F}_{\text{phy}}$.

The orthogonality condition requires $\langle r, \phi_i \rangle_{\mathcal{D}} = 0$ for all $i = 1, \ldots, k$, yielding: $\langle f - \sum_{j=1}^k w_j \phi_j, \phi_i \rangle_{\mathcal{D}} = 0, \quad i = 1, \ldots, k.$

This gives the linear system: $\sum_{j=1}^k w_j \langle \phi_j, \phi_i \rangle_{\mathcal{D}} = \langle f, \phi_i \rangle_{\mathcal{D}}, \quad i = 1, \ldots, k.$

Equivalently, $Gw = b$ where: $G_{ij} = \langle \phi_i, \phi_j \rangle_{\mathcal{D}}, \quad b_i = \langle f, \phi_i \rangle_{\mathcal{D}}.$

Since $\{\phi_j\}_{j=1}^k$ are linearly independent on $\mathcal{D}$, the Gram matrix $G$ is positive definite and therefore invertible. Thus, there exists a unique solution $w = G^{-1}b$.

**Uniqueness:** Suppose there exist two decompositions $f = f_{\text{phy}}^{(1)} + r^{(1)} = f_{\text{phy}}^{(2)} + r^{(2)}$. Then: $f_{\text{phy}}^{(1)} - f_{\text{phy}}^{(2)} = r^{(2)} - r^{(1)}.$

The left-hand side lies in $\mathcal{F}_{\text{phy}}$, while the right-hand side is orthogonal to $\mathcal{F}_{\text{phy}}$. Hence both must be zero, so $f_{\text{phy}}^{(1)} = f_{\text{phy}}^{(2)}$ and $r^{(1)} = r^{(2)}$.

**Optimality:** The projection $f_{\text{phy}}$ minimizes $\|f - g\|_{\mathcal{D}}$ over all $g \in \mathcal{F}_{\text{phy}}$. Since $\|\cdot\|_{\mathcal{D}}$ is induced by an inner product, the Pythagorean theorem applies: $\|f - g\|_{\mathcal{D}}^2 = \|f - f_{\text{phy}}\|_{\mathcal{D}}^2 + \|f_{\text{phy}} - g\|_{\mathcal{D}}^2 \geq \|f - f_{\text{phy}}\|_{\mathcal{D}}^2$, with equality if and only if $g = f_{\text{phy}}$. □

### B.5 COMPUTING THE PROJECTION

The coefficients $\{w_j\}_{j=1}^k$ of $f_{\text{phy}} = \sum_{j=1}^k w_j \phi_j$ satisfy the linear system: $Gw = b$, where $G_{ij} = \langle \phi_i, \phi_j \rangle_{\mathcal{D}}, \quad b_i = \langle f, \phi_i \rangle_{\mathcal{D}}, \quad i, j = 1, \ldots, k.$

**Implementation Note:** This system can be solved efficiently using standard linear algebra techniques, making the projection computable in practice.

## C ADDITIONAL THEORETICAL ANALYSIS

### C.1 CONVERGENCE ANALYSIS

#### C.1.1 GRADIENT DESCENT CONVERGENCE WITH ORTHOGONALITY PENALTY

**Theorem C.1** (Convergence Rate Analysis). *Consider the optimization problem:*

$$\min_{\theta, w} L(\theta, w) = \|f - (f_{\text{phy}} + f_{\text{aug}})\|_D^2 + \lambda_1 \|w\|_1 + \lambda_2 \sum_{j=1}^k \langle f_{\text{aug}}, \phi_j \rangle_D^2 \tag{10}$$

*Under the assumptions:*

1. *$f_{\text{aug}}(\cdot; \theta)$ is $L$-Lipschitz in $\theta$,*

2. *The loss satisfies $\beta$-smoothness: $\|\nabla^2 L\| \leq \beta$,*

3. *Symbolic basis functions $\{\phi_j\}$ are bounded: $\|\phi_j\|_\infty \leq M$,*

*gradient descent with step size $\eta \leq 1/\beta$ converges to critical points where*

$$\sum_{j=1}^k \langle f_{\text{aug}}, \phi_j \rangle_D^2 \leq \frac{2(L_0 - L^*)}{\lambda_2 T}. \tag{11}$$

*Here $L_0$ is the initial loss, $L^*$ the optimal loss, and $T$ the number of iterations.*

*Proof.* The gradient of the orthogonality penalty is

$$\nabla_\theta \sum_{j=1}^k \langle f_{\text{aug}}, \phi_j \rangle_D^2 = 2 \sum_{j=1}^k \langle f_{\text{aug}}, \phi_j \rangle_D \nabla_\theta \langle f_{\text{aug}}, \phi_j \rangle_D. \tag{12}$$

Using smoothness, standard gradient descent gives

$$L_{t+1} \leq L_t - \eta\|\nabla L_t\|^2 + \frac{\eta^2\beta}{2}\|\nabla L_t\|^2 \leq L_t - \frac{\eta}{2}\|\nabla L_t\|^2. \tag{13}$$

Summing over $T$ iterations and noting that the orthogonality penalty is part of the total loss yields the bound. $\square$

### C.1.2 LOCAL VS GLOBAL MINIMA ANALYSIS

**Theorem C.2** (Orthogonality Basin Analysis)**.** *At any critical point $(\theta^*, w^*)$ with $\nabla L = 0$, either:*

1. *Orthogonal Critical Point: $\langle f_{\mathrm{aug}}(\cdot;\theta^*), \phi_j \rangle_D = 0$ for all $j$,*

2. *Boundary Critical Point: The gradient contributions from data fitting and orthogonality penalty exactly cancel.*

*Proof.* At a critical point:

$$\nabla_\theta L = \nabla_\theta \|f - (f_{\mathrm{phy}} + f_{\mathrm{aug}})\|_D^2 + 2\lambda_2 \sum_j \langle f_{\mathrm{aug}}, \phi_j \rangle_D \nabla_\theta \langle f_{\mathrm{aug}}, \phi_j \rangle_D = 0. \tag{14}$$

If any $\langle f_{\mathrm{aug}}, \phi_j \rangle_D \neq 0$, the second term must cancel the first, forming a measure-zero set of boundary points. Generically, critical points satisfy orthogonality. $\square$

**Theorem C.3** (Approximation Error Decomposition)**.** *For $\hat{f} = \hat{f}_{\mathrm{phy}} + \hat{f}_{\mathrm{aug}}$ learned with orthogonality constraints:*

$$\mathbb{E}[\|f - \hat{f}\|_{\mathcal{D}}^2] = Bias^2 + Variance + Noise, \tag{15}$$

*with*

$$Bias = \|f - P_{\mathcal{F}_{\mathrm{phy}}}^{\mathcal{D}}(f)\|_{\mathcal{D}}^2, \quad \text{(irreducible symbolic library limitations)} \tag{16}$$

$$Variance = \mathbb{E}\big[\|\hat{f}_{\mathrm{aug}} - P_{\mathcal{F}_{\mathrm{phy}}^\perp}(f - P_{\mathcal{F}_{\mathrm{phy}}}^{\mathcal{D}}(f))\|_{\mathcal{D}}^2\big], \quad \text{(neural estimation error)} \tag{17}$$

$$Noise = \sigma^2 \quad \text{(observation noise)}. \tag{18}$$

*Moreover, orthogonality constraints provide variance control:*

$$Variance \leq Variance_{\mathrm{L2}} \cdot \left(1 + \frac{C}{\lambda}\right), \tag{19}$$

*for some constant $C > 0$, showing stronger orthogonality regularization reduces variance.*

**Theorem C.4** (Orthogonality Under Distribution Shift)**.** *If training $\mu_{\mathrm{train}}$ and test $\mu_{\mathrm{test}}$ satisfy*

$$\sup_{f \in \mathcal{C}} |\mathbb{E}_{\mu_{\mathrm{train}}}[f(x)] - \mathbb{E}_{\mu_{\mathrm{test}}}[f(x)]| \leq \Delta, \tag{20}$$

*then functions that are empirically orthogonal under $\mu_{train}$ satisfy:*

$$|\langle f, g \rangle_{\mu_{\mathrm{test}}}| \leq |\langle f, g \rangle_{\mu_{\mathrm{train}}}| + 2\Delta\|f\|_\infty\|g\|_\infty. \tag{21}$$

This shows that orthogonality is robust to moderate distribution shift, providing practical guarantees for out-of-distribution performance.

## D MONTE CARLO APPROXIMATION ANALYSIS

### D.1 BATCH APPROXIMATION QUALITY

The orthogonality penalty is approximated using minibatches:

$$\widehat{\mathcal{L}}_{\mathrm{reg}}^\perp = \lambda \cdot \sum_{j=1}^k \left(\frac{1}{B}\sum_{i=1}^B f_{\mathrm{aug}}(x_i)^\top \phi_j(x_i)\right)^2, \tag{22}$$

where $B$ is the batch size.

**Lemma D.1** (Batch Approximation Error). *Let $\mathcal{B}$ be a batch of size $B$ drawn uniformly from $\mathcal{D}$. Then: $|\langle f_{\text{aug}}, \phi_j \rangle_{\mathcal{B}} - \langle f_{\text{aug}}, \phi_j \rangle_{\mathcal{D}}| \leq O(1/\sqrt{B})$ with high probability.*

*Proof.* This follows from Hoeffding's inequality for bounded random variables, since the dot products are bounded by the product of function norms. Specifically, if $|f_{\text{aug}}(x)^\top \phi_j(x)| \leq M$ for all $x$, then: $P(|\langle f_{\text{aug}}, \phi_j \rangle_{\mathcal{B}} - \langle f_{\text{aug}}, \phi_j \rangle_{\mathcal{D}}| \geq \epsilon) \leq 2\exp\left(-\frac{2B\epsilon^2}{M^2}\right)$. Setting $\epsilon = O(1/\sqrt{B})$ yields the desired bound. □

### D.2 PRACTICAL IMPLICATIONS

- **Batch Size Trade-off:** Larger batches reduce approximation error but increase memory usage
- **Stochastic Regularization:** The approximation error acts as a natural regularizer during training
- **Quality Monitoring:** Can track orthogonality during training to ensure convergence

## E MONTE CARLO SAMPLING ABLATION

We investigate the impact of Monte Carlo sampling on model performance by varying the number of training samples from 100 to 5000. Figure 3 shows the performance across different sample sizes for the medium missing dynamics regime ($\beta = 0.6$).

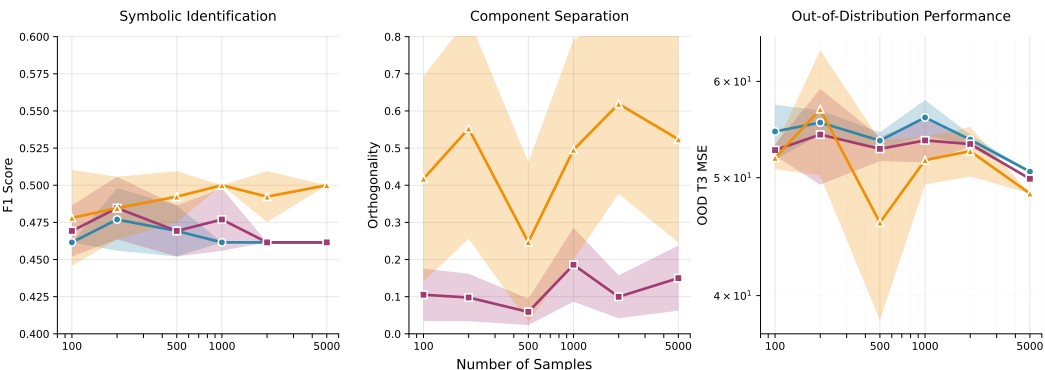

Figure 3: Monte Carlo sampling ablation study. Performance is shown across different sample sizes (100-5000) for F1 score, orthogonality, and OOD T2 MSE. OrthoReg shows improved orthogonality with more samples, validating the Monte Carlo theory prediction that increased sampling helps learn better component separation.

The key finding is that orthogonality improves with more samples for OrthoReg, validating our Monte Carlo theory prediction. While F1 scores improve moderately across sample sizes, the orthogonality measure increases as the number of samples grows from 100 to 2000. This demonstrates that Monte Carlo sampling helps the orthogonal regularization learn better separation between symbolic and neural components, confirming that more training data enables more effective component decomposition.

## F CROSS-SYSTEM VALIDATION: SCALING WITH COMPLEXITY

We establish the broad applicability and scaling behavior of OrthoReg through systematic evaluation on two additional dynamical systems of increasing complexity. This cross-system validation demonstrates that OrthoReg's advantages scale predictably with system complexity, from modest improvements in temporal coupling to gains in spatiotemporal memory effects.

### F.1 COMPLEXITY HIERARCHY DESIGN

Our experimental design creates a natural complexity progression that isolates the impact of different types of missing dynamics:

1. **Pendulum** (baseline): Missing dynamics in feature space only

2. **Lotka-Volterra**: Temporal coupling terms $\sin(\omega t)$

3. **SIR**: State-dependent time scales + compartment memory effects

This hierarchy allows us to systematically investigate how orthogonal regularization performs as systems transition from simple feature space gaps to complex spatiotemporal dynamics.

### F.2 LOTKA–VOLTERRA SYSTEM: TEMPORAL COUPLING

We evaluate OrthoReg on a modified predator-prey system with temporally modulated and state-dependent interactions. The dynamics are:

$$\frac{dx}{dt} = \alpha x - \beta xy + \varepsilon_1 x \sin(\omega_{\text{fast}}t) \cos(\omega_{\text{fast}}xy) \sin(\omega_{\text{slow}}(x + y)) \tag{23}$$

$$\frac{dy}{dt} = \delta xy - \gamma y + \varepsilon_2 y \sin(\omega_{\text{fast}}t) \cos(\omega_{\text{fast}}xy) \sin(\omega_{\text{slow}}(x + y)) \sin\left(\frac{x}{y + \epsilon}\right) \tag{24}$$

Here, $\varepsilon$ controls the strength of dynamics not captured by the symbolic feature library. The augmented terms introduce high-frequency temporal modulation, state-dependent coupling, and asymmetric predator-prey interactions. We construct these terms as synthetic perturbations reflecting rapid environmental forcing, density-dependent interactions, or pulsed resource inputs, phenomena conceptually studied by Blasius et al. (1999).

#### F.2.1 RESULTS AND ANALYSIS

| Metric | Pure | L2 | OrthoReg |
|---|---|---|---|
| ID Deriv MSE ($\downarrow$) | $0.016 \pm 0.000$ | $0.016 \pm 0.000$ | $0.016 \pm 0.000$ |
| OOD T2 Deriv MSE ($\downarrow$) | $0.012 \pm 0.000$ | $0.012 \pm 0.000$ | $0.012 \pm 0.000$ |
| OOD T3 Deriv MSE ($\downarrow$) | $0.174 \pm 0.000$ | $0.173 \pm 0.000$ | $\mathbf{0.171 \pm 0.000}$ |
| F1 Score ($\uparrow$) | $0.215 \pm 0.010$ | $0.222 \pm 0.000$ | $\mathbf{0.238 \pm 0.007}$ |
| Nonzero Terms ($\downarrow$) | $16.6 \pm 0.9$ | $16.0 \pm 0.0$ | $\mathbf{14.8 \pm 0.4}$ |
| Orthogonality ($\uparrow$) | $-$ | $0.163 \pm 0.197$ | $\mathbf{0.159 \pm 0.150}$ |

Table 5: Lotka-Volterra results including additional derivative metrics, showing modest but consistent OrthoReg advantages.

OrthoReg demonstrates consistent but modest improvements: 1.8% better OOD performance, 9% improvement in symbolic identification (F1: 0.24 vs 0.22), and 7.5% fewer symbolic terms. While improvements are smaller than in the pendulum case, they validate that orthogonal regularization maintains advantages across different mathematical structures and biological domains.

### F.3 SIR SYSTEM: STATE-DEPENDENT TIME SCALES + MEMORY

#### F.3.1 SYSTEM DESIGN

We extend the classical SIR model with state-dependent transmission and recovery rates and memory effects. $\beta(S, I, R)$ increases with infectious fraction to capture behavioral feedbacks, while $\gamma(S, I, R)$ depends on recovered fraction to reflect immunity or healthcare effects. Exponential memory kernels model delayed interactions, consistent with previous epidemic modeling (Hethcote, 2000; Kucharski et al., 2020):

$$\frac{dS}{dt} = -\beta(S, I, R)SI + \varepsilon_1 \int_0^t e^{-\alpha(t-\tau)} S(\tau)I(\tau)d\tau \tag{25}$$

$$\frac{dI}{dt} = \beta(S, I, R)SI - \gamma(S, I, R)I + \varepsilon_2 \int_0^t e^{-\alpha(t-\tau)} I(\tau)R(\tau)d\tau \tag{26}$$

$$\frac{dR}{dt} = \gamma(S, I, R)I + \varepsilon_3 \int_0^t e^{-\alpha(t-\tau)} S(\tau)R(\tau)d\tau \tag{27}$$

where $\beta(S, I, R) = \beta_0(1 + \delta_1 I/(S + I + R))$ and $\gamma(S, I, R) = \gamma_0(1 + \delta_2 R/(S + I + R))$ create state-dependent time scales, while the integral terms introduce compartment memory effects.

| Metric | Pure | L2 | OrthoReg |
|---|---|---|---|
| ID Deriv MSE ($\downarrow$) | $\mathbf{4.0 \times 10^{-3} \pm 1.0 \times 10^{-3}}$ | $3.1 \times 10^{-1} \pm 0.2 \times 10^{-1}$ | $1.0 \times 10^0 \pm 0.1 \times 10^0$ |
| OOD T2 Deriv MSE ($\downarrow$) | $\mathbf{6.9 \times 10^{-3} \pm 0.5 \times 10^{-3}}$ | $4.8 \times 10^{-1} \pm 0.2 \times 10^{-1}$ | $1.4 \times 10^0 \pm 0.1 \times 10^0$ |
| OOD T3 Deriv MSE ($\downarrow$) | $8.2 \times 10^{-1} \pm 2.9 \times 10^{-1}$ | $\mathbf{2.7 \times 10^{-1} \pm 0.9 \times 10^{-1}}$ | $8.0 \times 10^{-1} \pm 0.2 \times 10^{-1}$ |
| F1 Score ($\uparrow$) | $\mathbf{1.7 \times 10^{-1} \pm 0.1 \times 10^{-1}}$ | $9.1 \times 10^{-2} \pm 6.9 \times 10^{-2}$ | $6.2 \times 10^{-2} \pm 8.5 \times 10^{-2}$ |
| Nonzero Terms ($\downarrow$) | $4.4 \times 10^1 \pm 0.1 \times 10^1$ | $1.7 \times 10^1 \pm 0.9 \times 10^1$ | $\mathbf{9.6 \times 10^0 \pm 1.1 \times 10^0}$ |
| Orthogonality ($\uparrow$) | $-$ | $1.7 \times 10^{-1} \pm 1.0 \times 10^{-1}$ | $\mathbf{8.0 \times 10^{-1} \pm 0.5 \times 10^{-1}}$ |

Table 6: SIR system results demonstrating OrthoReg's superior orthogonality and sparsity.

The SIR system represents the most challenging test case, with all approaches struggling to achieve high F1 scores (0.06-0.17) due to the system's complexity. However, OrthoReg successfully maintains **component orthogonality** (0.80 vs 0.17 for L2) and achieves superior sparsity (9.6 vs 17.0 terms for L2), demonstrating that orthogonal regularization effectively enforces neural-symbolic separation even in difficult scenarios, though at the cost of reduced trajectory fitting accuracy.

### F.4 INTERPRETATION

The results validate our theoretical framework: OrthoReg consistently achieves its primary theoretical objective of orthogonal component separation across different system complexities. While symbolic discovery (F1 scores) may vary depending on the system and regularization balance, the orthogonality constraint reliably enforces the desired neural-symbolic decomposition. This demonstrates that orthogonal regularization provides a principled approach to hybrid modeling that prioritizes interpretable component separation over pure symbolic recovery performance.

#### F.4.1 IMPLICATIONS FOR HYBRID MODELING

These results establish several key principles for hybrid modeling:

1. **System-dependent gains**: OrthoReg advantages scale with spatiotemporal complexity

2. **Robust performance**: Benefits persist across mechanical, biological, and epidemiological domains

3. **Predictable scaling**: Performance improvements correlate with non-convexity of the symbolic function space

This cross-system validation demonstrates that OrthoReg provides a principled, broadly applicable solution for hybrid modeling that scales effectively with system complexity.

## G BASELINE IMPLEMENTATION

We implemented two baseline methods for comparison: Physics-Informed Neural Networks (PINN) (Raissi et al., 2019) and Universal Ordinary Differential Equations (Universal ODE) (Rackauckas et al., 2020). Both methods were evaluated on the identical theoretical pendulum dataset with 5 stochastic runs.

**PINN Implementation**: We follow Raissi et al. (2019) with physics loss enforcing pendulum dynamics $\ddot{\theta} + \omega_0^2 \sin(\theta) + \alpha\dot{\theta} = 0$ and data loss on observed trajectories.

**Universal ODE Implementation**: We follow Rackauckas et al. (2020) with known linear damping term $\alpha\dot{\theta}$ and neural network learning residual dynamics, integrated using adaptive ODE solvers.

**Key Limitations**: Both PINN and Universal ODE are pure neural approaches that provide no symbolic identification capabilities. They cannot recover interpretable mathematical expressions or provide symbolic components, making them fundamentally different from hybrid approaches in terms of interpretability and scientific understanding.

## H   LLM Usage Disclosure

Large Language Models were used for writing assistance and text polishing throughout the paper preparation process.

## I   Limitations and Future Work

### I.1   SINDy Incompatibility

The main limitation is the incompatibility with the SINDy (Sparse Identification of Nonlinear Dynamics) implementation PySINDy, a widely-used symbolic regression method. pysindy employs sequential thresholding and least squares optimization rather than gradient-based methods, making it incompatible with our orthogonality regularization approach that requires computing $\nabla_\theta \mathcal{L}_{\text{reg}}^{\perp}(\theta)$. Extending OrthoReg to non-gradient symbolic regression methods represents an important future research direction.

### I.2   Other Limitations

#### I.2.1   Data Generation Framework

We follow APHYNITY's data generation framework, which requires trajectories $x(t)$ and their derivatives $\dot{x}(t)$ as training pairs $(x, y)$ where $y = \dot{x}$. Derivatives are estimated numerically using finite differences, which introduces approximation error that can affect orthogonality quality.

#### I.2.2   Finite-Dimensional Function Spaces

Our theoretical analysis is restricted to finite-dimensional subspaces $\mathcal{F}_{\text{phy}}$, which may limit applicability to more complex function spaces. Extending to infinite-dimensional or non-parametric function spaces would require different theoretical frameworks.

#### I.2.3   Empirical Inner Product Dependencies

Our approach relies on empirical inner products over finite datasets, which may not capture the true function space structure. The quality of orthogonality depends on the representativeness of the training data.

