# OpenReview forum: "Hybrid Symbolic-Neural Models for Dynamical Systems"
_ICLR.cc/2026/Conference — ICLR 2026 Conference Withdrawn Submission_

### Official Review · Reviewer_ZZUw · 2025-10-29

**Soundness:** 1
**Presentation:** 1
**Contribution:** 2
**Rating:** 0
**Confidence:** 4

**Summary:**

This paper addresses the challenge of learning non-overlapping representations in hybrid modelling. The authors argue that standard L2 regularization is insufficient to prevent the neural component from learning dynamics that could be (partially) captured also by the symbolic part. In particular, they demonstrate that this arises when the symbolic component is learned together with the neural part.
Therefore, they propose Orthoreg, a regularization method that enforces orthogonality between the components. In practice, this means computing the product of the two components over all data points, and adding this term to the loss.
They provide a theoretical analysis and empirical evaluation of this approach in limited settings with different evaluation metrics.

Although the idea is interesting for hybrid modelling and theoretically justified, the paper's informal writing, lack of code availability, limited novelty with respect to a broader literature, and limited experimental evaluations make this work unsatisfactory with respect to the standards required by the conference.

**Strengths:**

This paper addresses a clear problem of hybrid modelling, and clearly shows why L2-based methods like APHYNITY do not learn orthogonal representations in the non-convex scenario of sparse symbolic regression. The theoretical approach is well-founded and deserves more attention. On the other hand, the experimental results show the effectiveness of this approach on multiple metrics (also OOD and an orthogonality score), hence going beyond pure MSE.

**Weaknesses:**

- Code is missing, hence accountability of the results is hindered.
- Baselines with direct related works ((Py)SINDy and APHYNITY) are missing.
- Writing and grammar could be improved substantially: there are some typos, informal expressions (reported between quotation marks "..."), bullet point missing capital letters and columns (e.g., in the introduction), "L2" written also as $L^2$, missing space in ODE definition, the word "dissemination" is used improperly (maybe meaning dissection?), "Pythagorean theorem" should be replaced with at least triangle inequality, ... overall, the tone should be more suited for an academic top-tier conference.
- Some important limitations are relegated to Appendices, and deserve more prominence in the main text. E.g., the use of gradient-based optimization and its restrictions with PySINDy.

**Questions:**

- While the paper's focus on dynamical systems and hybrid models is clear, the core idea of enforcing orthogonality and having additive components is related to broader concepts in statistics and ML (starting from PCA and its variants, or Generalized Additive Models and their neural variants, see also Additive Gaussian Processes Revisited (Lu 2022). A brief discussion situating OrthoReg within this wider literature could strengthen the paper's contribution and clarify its relationship to other forms of model decomposition.
- The finite-sample guarantees are not obvious. How does it derive from Hoeffding's inequality?
- Implementation details are poorly described. How is the data generated? How are the baselines implemented, trained and fine-tuned?
- What is the symbolic library you employed? How did you learn its parameters?
- Could you report the distinct performances of Orthoreg but without the f_aug component? In such an ablation, it would be clear how much f_aug contributes to the final scores.
- The paper states a "modest 5-15% computational overhead." Could you be more specific?
- How was the crucial orthogonality regularization strength lambda selected for the main experiments in Tables 1, 2, and 3? Figure 2 provides an excellent ablation on its effect, but it is not clear if lambda was tuned on a validation set for the main comparisons or if a fixed value was used.

---

### Official Review · Reviewer_pN8V · 2025-10-30

**Soundness:** 2
**Presentation:** 3
**Contribution:** 1
**Rating:** 2
**Confidence:** 4

**Summary:**

This paper proposes a new approach to learn dynamical systems modelled as autonomous ODEs, with two components: a  symbolic one and a neural net one. The goal is for the neural net part to act as a correction to a base symbolic model, given its high expressive power. The authors propose a novel formulation for the loss function that ensure that the base symbolic component and the neural correction are orthogonal, hence avoiding overlap between the two components. The approach was tested on synthetic data of damped
pendulum, Lotka–Volterra predator-prey system, and a memory-modulated SIR epidemiological
model.

**Strengths:**

- The proposed formulation to ensure orthogonality is novel despite its simplicity
- The paper is written clearly
- Evaluation on OOD data is interesting

**Weaknesses:**

- Limited evaluation: synthetic data only
- It seems the authors don't clearly define what they mean by OOD
- Relevance of hybrid modelling for practical problems is not clear
- Limited review of related work: recent works [1, 2, 3, 4] show that purely symbolic models are highly expressive and can even beat neural network-based models in certain settings. What symbolic models mostly lack is scalability. Discussion on these related works would be relevant and necessary.

[1] Sun, Fangzheng, et al. "Symbolic physics learner: Discovering governing equations via monte carlo tree search." arXiv preprint arXiv:2205.13134 (2022).

[2]  Dakhmouche, Ramzi, Ivan Lunati, and Hossein Gorji. "Robust Symbolic Regression for Dynamical System Identification." Transactions on Machine Learning Research. 2025.

[3] Liang, Senwei, and Haizhao Yang. "Finite expression method for solving high-dimensional partial differential equations." Journal of Machine Learning Research 26.138 (2025).

[4] Qian, Zhaozhi, Krzysztof Kacprzyk, and Mihaela van der Schaar. "D-code: Discovering closed-form odes from observed trajectories." International Conference on Learning Representations. 2022.

**Questions:**

1/ Could you clarify what you call L2-reg in section 5.1 ?

2/ Could you give an example of a practical problem where hybrid modelling leads to considerable insights not accessible with purely neural/symbolic modeling ?

3/ What's the sampling frequency for each tested dynamical system ?

---

### Official Review · Reviewer_e6cB · 2025-10-31

**Soundness:** 3
**Presentation:** 3
**Contribution:** 2
**Rating:** 4
**Confidence:** 3

**Summary:**

Manually specified dynamical systems are interpretable but simplistic and data-driven models are not physical. The paper combines physics-models with neural networks in a hybrid approach.The problem is that the data-driven component may learn the mechanistic parts more appropriately modeled by the physics model. The function is modeled as f = fp + fa where fp and fa are the physical (learned via basis functions) and augmented parts (a NN) of the model. When the symbolic structure is fixed the components can be separated by L2 regularizing the augmented model. However when the symbolic and augmented parts are learned jointly this is no longer the case and the optimization is non-convex.

 Standard L2 regularization is not enough and this paper introduces a regularization method that enforces orthogonality between the symbolic and mechanistic parts. The resulting model improves OOD generalization, symbolic identification and sparsity.

The method is tested on three benchmark datasets: damped pendulum, Lokta Volterra and an SIR system showing improved structural recovery performance.

**Strengths:**

The paper is well-structured and clear. It explains the prior work, identifies a clear gap in the prior work (the APHYNITY model and analysis), generalizing to the case  where the structure and augmentation are jointly learned and proposes a solution.

The authors provide a theoretical analysis of the approximation error from quadratic penalty theory.

The method achieves higher symbolic recovery than the baselines.

The method also shows higher out of distribution performance.

**Weaknesses:**

The proposed method itself is not very novel. The solution explicitly regularizes the augmentation function to be orthogonal to the basis functions with L2 regularization. The regularization strength is increased as in the penalty method to achieve a strict constraint.

The experiments appear to be all done on small scale systems with synthetic data. From what I can see, there is no study of the effect of noise on the results and the experiments are performed on synthetic data. The baselines that are compared with are also not recent.

**Questions:**

What are the sizes and dimensions of the systems that you compare with? Have you tried larger systems than those with only a few dimensions?

Why are UDEs unable to recover symbolic components (line 274)? Can’t they include symbolic terms in addition to neural networks?

---

### Official Review · Reviewer_EjE5 · 2025-11-04

**Soundness:** 2
**Presentation:** 2
**Contribution:** 2
**Rating:** 2
**Confidence:** 4

**Summary:**

This paper introduces OrthoReg (Orthogonal Regularization) to address redundancy and uninterpretability in hybrid approaches for dynamical systems modeling, where the neural network often relearns dynamics already captured by a symbolic, physics-based component, instead of only modeling the residual. OrthoReg encourages orthogonality between the two components using a regularization term. Results on benchmark systems show that OrthoReg improves out-of-distribution generalization, enhances symbolic identification, and increases model sparsity, overall improving hybrid modeling performance and interpretability.

**Strengths:**

1. The approach is very simple and backed up by theoretical rigor.
2. The method is applicable to many hybrid-symbolic approaches, which makes it useful for the wider SciML community.
3. The paper is mostly easy to follow, but lacks some details on experimental design (see Weaknesses).

**Weaknesses:**

1. The paper has a severe lack of details on experimental design. It does not mention what exactly some of the metrics compute. E.g., how exactly are in-distribution, OOD–T2 and OOD-T3 evaluated? How is data sampled from the benchmark systems? What is the exact architecture of the NN architecture? etc.
2. The paper does not contain any example trajectories of the benchmark systems and/or fitted models. Including those would help the reader get a qualitative feel of the problem and model performance besides the employed quantitative measures and descriptions of the data (e.g. “The augmented terms introduce high-frequency temporal modulation, state-dependent coupling [...]” in Appx. F.2).
3. Moreover it is not clear how well the learned model (symbolic + residual) captures long-term dynamics ("climate") of the benchmark systems, i.e. does the stability of the model reflect the data? If not, is does the approach approve over the other baselines in the paper?
4. There is a lack of more complex dynamical phenomena, e.g. benchmark systems that exhibit chaos or multistability. Moreover, the method is not tested on any real-world data.

Minor stuff:
- Fig. 3 lacks a legend
- Paragraphs in ll. 447 - 457 contain the same information and seem redundant, probably the authors forgot to comment out one of them?
- l. 475: subscripts seem misformatted

**Questions:**

1. "The orthogonality computation requires $\mathcal{O}(kBd)$ operations per forward pass with modest 5-15% computational overhead" - How is this overhead determined?
2. How are low/medium/high missing dynamics regimes determined/set? Are these ad-hoc values or is there specific reason for these exact values/scales?
3. How exactly is “irregular sampling” performed? Why does it deteriorate derivative errors? Do the others estimate the vector field from data first? (see Weaknesses: limited experimental design explanation)
4. A big problem in dynamical systems modeling is out-of-domain generalization in face of multistability [1]. Can OrthoReg help in finding solutions that generalize to unobserved basins of attraction when the library of the symbolic compartment is underspecified (see e.g. Duffing system in Fig. 2b in [1]). Would be great to see results on this!
5. Orthogonality seems to exhibit high variance between runs for increasing $\lambda$ (Fig. 2). Can the authors comment on this?

**References**:

[1] Göring, Niclas, et al. "Out-of-domain generalization in dynamical systems reconstruction." Proceedings of the 41st International Conference on Machine Learning. 2024.

---

### Note · Authors · 2025-11-24

**Comment:**

We thank the reviewers and the AC for the constructive feedback and the time invested in evaluating our submission. Overall, the reviews were positive about the general idea behind our methodology and the rather negative scores appear to be mostly due to fixable aspects. Therefore, we will withdraw the submission and prepare a substantially revised version that incorporates all suggestions and plan to submit it to another venue.

**Withdrawal Confirmation:**

I have read and agree with the venue's withdrawal policy on behalf of myself and my co-authors.